# Once upon a Pandemic: ‘Online’ Therapeutic Groupwork for Infants and Mothers Impacted by Family Violence

**DOI:** 10.3390/ijerph192316143

**Published:** 2022-12-02

**Authors:** Wendy Bunston, Dianne J. Haufe, Jillian R. Wallis, Robyn Fletcher, Adrian J. Mether

**Affiliations:** 1Wb Training & Consultancy, Moonee Ponds, VIC 3039, Australia; 2Department of Community and Clinical Health, La Trobe University, Bundoora, VIC 3086, Australia; 3Uniting Care Wesley Bowden, Bowden, SA 5007, Australia; 4Mind Australia, Mile End, SA 5031, Australia; 5Berry Street, Take Two—Restoring Childhood, Eaglemont, VIC 3084, Australia

**Keywords:** infant mental health, domestic and family violence, groupwork, online therapy, COVID 19 pandemic

## Abstract

This case study describes the transition to an ‘online’ delivery of an evaluated infant mental health group work intervention for infants/mothers impacted by family violence during the COVID 19 pandemic. The imperative to provide early intervention to infants and their mother is outlined. The model and practice principles integral to this intervention are provided and described are four separate groups run online within two different Australian cities. Facilitators of the groups found that they were able to hold the infants and mothers safely in the online space despite the unexpected presence of others in the families’ homes. The home-based nature of the work caused by lockdown restrictions revealed a transparency not found in office-based work, whilst simultaneously, evoking some discomfort. The ease with which infants and young children embraced technology worked in favor of using the online space. Playful, restorative, and creative ways of engaging with a highly vulnerable cohort of families were achieved; enhancing relational repair following both family violence and the isolation created by restrictions imposed by lockdowns. Despite an initial hesitancy to move online, the authors discovered meaningful ways through which to engage, treat and provide safe relational repair work with infants and their mothers impacted by family violence.

## 1. Introduction

Infants and young children under five years of age are more often present during, as well detrimentally impacted by, exposure to family violence than any other age group in childhood. However, they are least likely to receive any adequate service response or recognition within the literature [1,2]. Within Australian data, Aboriginal and Torres Strait Islander children are overrepresented in their exposure to and the impacts of Domestic and Family Violence (DFV), are more likely to be hospitalized [3] and experience homelessness [4].

Significant and repeated trauma experiences in the early years lays down an implicit, preverbal and physiological foundation which has the potential to operate throughout the individual’s entire life [5]. The infant exposed to the toxicity of relational violence, often intergenerational, bleeds into every developmental crevice, every sensate expression, every memory, non-declarative; and if sustained, declarative [6,7].

The imperative for specialist early intervention groups to address the impacts of familial violence on infants and young children is gaining traction within the DFV sector [8]. Prior to 2020, such infant mental health informed work was optimally delivered through direct face-to-face therapeutic work. However, the landscape for delivering therapeutic services to vulnerable families changed dramatically in March 2020. As the COVID-19 pandemic hemorrhaged throughout the world, DFV and social work and specialist family services were required to quickly adapt. This adaptation was not in relation to their commitment to delivering sound psychological and practical support but in how this could be safely and effectively done.

This practiced based paper explores the themes and learnings to emerge through the online delivery of four distinct infant/mother groups run by two different services in two different Australian states during COVID-19 lockdown periods. Beginning with a brief outline of the Infant/Mother Peek a Boo ClubTM (PABC) intervention, the adaptation of this groupwork intervention to an online format will be described, and how social support services from two different states during Australia’s lockdown learnt from one another; as well as from the infants and mothers themselves. Further explored is the use of online technology to deliver therapeutic support, the persistent exposure of very young children to screen time, and a curiosity about how children process their experiences of interactive technology. Further explored is and how we, as service providers, may enter the online world with vulnerable families and provide opportunities to grow healthy relational opportunities post family violence. The paper will conclude with how these learnings, emerging from this unusual period in our global history inspired inventiveness whilst retaining relational continuity and safety for highly vulnerable families.

### 1.1. The Need to Treat Infants and Mothers Impacted by Family Violence

The imperative to treat infants impacted by family violence has been informed by research which demonstrates that such maltreatment is linked to an increased likelihood of anxiety and depression during adolescence [9], increased risk of cancer, and a reduction in healthy immune systems and cardiovascular functioning in later life [10]. Attention and cognitive deficits, poor verbal memory retrieval, and diminished cerebral volume have also been reported [11]. Included in the 17,337 adults involved in The Adverse Childhood Experiences (ACE) study were a significant number who reported early experiences of DFV [12]. A correlation was found between early adversity and numerous health and relational difficulties, including substance abuse and neuropsychiatric conditions.

### 1.2. Groupwork for Infants and Mothers Impacted by Family Violence

The Peek a Boo ClubTM was developed in 2005 [13,14] and run-in multiple regions within Victoria, Australia, and more recently South Australia. The PABC is an evaluated, specialist infant mental health therapeutic group work intervention for infants and their mothers impacted by family violence [14]. Infants up to age 5 are the focus of the intervention and considered equal participants in the group. Thus, a group with 8 participants could be four infants and four mothers. The duration of the group is generally between 8 to 10 weeks.

Critical to the intervention is the use of an ‘infant led’ approach [13,15,16] which makes prominent the subjectivity of the infant and fosters the capacity of the adults in the group to reflect on the infant’s experience; both in the here and now of the group, and in mothers empathically connecting with their own past, early childhood experiences [17,18]. Intimate trauma such as family violence has often been experienced intergenerationally. Relational violence creates ruptures and/or the prevention of relational availability within the still forming infant/caregiver relationship to defend against a threat to life (emotional, mental, and physical). Such threats or the profound lack of experiencing a sense of safety leads to obnoxious impingements on the internal formation of the early sense of self, and self in relation to others [19,20]. The intervention is guided by practice principles which create gentle and safe relational exploration and repair of what the infant and their caregiver brings into the intimate space of the group, rather than a strict adherence to a rigid model of delivery. This is to remain congruent to an approach which honors the rapidly emerging and unique subjectivity of the infant [20,21,22]. Ideally, the infant’s subjectivity is nurtured, developing within the crucial, life giving and relationally sustaining intersubjective relationship between the infant and their primary caregiving system. The PABC intervention works to proactively grow the infants experience of self, safety, and mastery in relation to others, and most importantly that of their primary caregiver.

The PABC intervention itself has been described in detail in other publications [13,14,23] and was developed as an extension of an existing program addressing the impacts of family violence on children [24,25,26], instead focusing on infants and their caregivers [13]. An extensive evaluation of the model was undertaken with mothers and their infants who participated in the group work intervention from 2007–2011 (n-128). The demographic data collected for this evaluation demonstrated that over 90% of women participating had experienced more than two types of violence, with well over 70% of the perpetrators of the violence being the father of the infant. Just under half had child protection involvement and nearly a third of the mothers reported suffering from PTSD. Well over half the mothers identified their nationality as Australian [14]. The evaluation of the intervention demonstrated that “the PABC intervention was associated with improved scores on outcome measures assessing infant, mother and infant–mother functioning…infants appeared to have developed new ways of regulating and modulating their behaviors, whilst mothers were more attuned and available to assist the infants to contain strong emotional responses” ([14], p. 129). While the COVID-19 pandemic has forced seismic shifts in how the therapeutic world delivers services to their clients, the principles of practice underpinning this targeted intervention remained constant.

### 1.3. The PABC Practice Principles

The PABC model of practice is both organic and steadfast in its approach. This work calls for the ability to explore, discover and grow relational opportunities for each unique individual who makes up each unique group. Non-negotiable is the stability of the scaffold upon which this organic work grows. This involves remaining committed to practice principles which involve:Being Infant Led, which affords the infant the right to be seen, directly engaged with, spoken to and ensures that their experience is considered [13,22].Undertaking a thorough initial assessment session, ensuring that the infant is present, and where the tone is set in demonstrating what is infant led practice, privileging the engagement of both infant and mother/caregiver.An adequate number of facilitators involved in running this intense, process orientated group. Ideally this requires a minimum of two but benefits from more.Appropriate group size should ideally be eight participants but not exceed ten group participants. The infant is an equal participant in the group and counted as such.Adequate time to complete all aspects of the practice model’s intervention, ideally, a complete day.Reflective supervision occurs weekly. Imperative to therapeutic work addressing family violence is ensuring a reflective space with an experienced supervisor [27].The provision of a safe, welcoming, protected, and consistent space within which to hold the group each week, with access to the same toys, sitting on floor cushions and with the same facilitation team each week.Provision of core rituals, providing a consistent frame for the group experience itself, includes a welcome ritual, a midway break for morning tea and a closing ritual. This is then followed by sending participants a weekly therapeutic newsletter.Set a clear time frame for the group, ideally eight weeks.Managing closure and marking this with an additional reunion session held four to six weeks post the groups conclusion.Reciprocal feedback sessions take place post group (and prior to the group reunion) with each dyad (or triad if twins or siblings attend).

These practice principles act as the anchor for a dynamic as well as organic therapeutic process which unfolds and builds week after week and is exclusive to each group.

### 1.4. The PABC Model, Goals and Objectives

The PABC group work intervention is designed for infants from 0–4 years, and their mothers (or carers) who have experienced DFV. It is not manualised, but process driven. It delivers an organic, experiential, interactive format that creates space for the infant ‘to be’ in relationship with others. It privileges the experience of the infant and supports the infant and mother/carer to form and consolidate healthier attachment patterns. Generally, groups have between 8 to 12 participants (the infants are counted as a participant in their own right) and 2 to 4 Infant Mental Health or Allied Health Facilitators who are trained in the model [13,28].

The group runs for 2 h, is generally held in community locations, and commences with a pre-individual (infant/mother) intensive, pre-group assessment session. Immediately after each group finishes, facilitators write up infant observation process notes [29,30]. These notes help facilitators collectively distilled the minutia of the multiple relational interactions that occur between infant and mother, and all other group members. This is followed by facilitators attending reflective supervision [27,31] and further distilling the major insights, learnings and unconscious material and re-enactments that arise throughout each unique group session, each week. This is followed by facilitators collectively writing up a therapeutic newsletter to send out to each mother or carer before the next group. This newsletter encapsulates the essence of what has occurred within that session, reflecting on what the infants have shown us about their experience of their mothers, each other, and their experiences of DFV. Following the final week of the PABC intervention infant/mother reciprocal feedback sessions are arranged. Two months post the final group session a reunion is held.

The objective of the intervention is to bring the subjectivity of the infant alive in the mind, actions, and relational world of the mother. This includes recognition of the feeling states of the infant impacted by DFV, as well as recognition of their own (the mother’s) trauma responses. Be this from their own early childhood and/or from their own experience of DFV. How this plays itself out in the present is reflected on, and how they may see this being transformed in their future. Bringing the infant to the fore; reflecting on the infants’ experiences in the here and now of the group [32]; recognising the attachments the mother has experienced in her life and reflecting on the attachment now forming between the infant and their mother offers these infants and mothers a powerful healing experience. The goal of the group is to create a safe space to unpack the enormity of the trauma they have survived and to fan the hope the infant carries for mothers of creating a very different relational future [22].

## 2. Case Description

One service had just commenced forming a group when the pandemic forced a lockdown in their state. The other service began the entire process of recruitment and group formation during the lockdown period. That the first service in one state had adapted quickly to an online space encouraged the other service to forge ahead into this unexplored virtual delivery space.

### 2.1. Service One—Group Online

Four infants and three mothers participated in three, separate, home based assessment sessions. All mothers were born outside of Australia and two spoke English as a second language. Week one of the group took place in-person. The following week, across Australia, the country was plunged into the chaos created by the pandemic. Workers were ordered, where possible, to work from home. The facilitators decided to provide identical activity packs to each family, delivering these to their front door. Each pack contained an initial newsletter with instructions on how to access the group online; a summary of the first session and information about what to expect each week during lockdown; and 3 weeks of activities.

Despite the initial concern that the absence of coming together in the one, contained space, would hamper forming as a group, unexpected virtual opportunities emerged. Facilitators quickly adapted to singing online using a ‘hello song’ to signal the beginning of each session. Mothers would either sit with their infant or follow them through their home with mobile phone or iPad in hand to ensure they remained in view. One mother gained enough confidence to happily sing her child’s favorite songs during group, enabling the rest of the group began to join in.

When toddlers wandered off to amuse themselves, or mothers become engrossed in conversations with one another, facilitators kept the infant in mind, gently sharing observations and reflecting on what they saw in the group space to keep the child from being left unheard, their needs unrecognized or disappeared. In the home space facilitators were given an insight into the everyday life of the child and their mother. One mother often remained in bed with the child at the beginning of the session. Another child often sat repeatedly watching his favorite you tube clip on an iPad which another child in the group recognized, calling out the name of the character in the clip. The nuances of what were seen, heard, and felt in the exchanges on screen all contributed to gaining insights into the relational world of the very young children participating in the group.

With the group occurring in their homes, much was revealed about how the mothers and children operated in their own space. One mother set up a special rug, toys and the activities sent to their home in readiness for the session each week. On occasion older siblings were present in the background adding understanding to the richness of the infant or young child’s relational world. A family would join the group late allowing more one on one time with a particular family who logged in earlier. One mother would turn on her sound but not her camera. The noises heard suggested that both mother and child were getting themselves ready so they could hear and talk to the rest of the group before appearing.

Additional anxieties were triggered by the pandemic. This group was multiracial and cut-off from supports both within Australia and overseas. Concern about the welfare of family members who lived in other countries significantly impacted by the pandemic were ever-present. Longstanding issues of grief and loss were triggered. The continuity of the facilitator’s availability each week online, and the consistent arrival of the therapeutic newsletter three days later, kept these three vulnerable families anchored, albeit in a virtual space.

### 2.2. Service One—Group Two Online

As health, social and welfare workers remained home based so too did vulnerable families, with new referrals dropping dramatically. Despite data from front line emergency services showing a decrease in reporting DFV and accessing emergency services [33] practitioners worried that enforced isolation during the pandemic was placing some infants, children and parents at further risk [34,35]. Facilitators invited the families from their first online group back for another term, hoping to consolidate the changes already occurring and looked for markers of change between the ending of the first group and the commencement of the second group.

The mother who had struggled to log on, on time, each week in the previous group generally arrive on time but still in bed with her toddler snuggled up asleep beside her. The television in their room often remained on during the online session, hindering the young child’s ability to fully participate in the online group, and creating the dilemma of which screen to focus on. However, this young child demonstrated a marked increase in participation in this second group, increasingly interacting with facilitators and other group members online, and engaging in impromptu singing.

The continuity these weekly sessions offered saw a slow blossoming of relational curiosity with previous group conversations referred to, exchanges of observations about each other’s children, and simultaneously, a noticeable shift in focus. As can occur in face-to-face groups, mothers can see the PABC as their time to talk with to other mothers. This resulted in one mother double booking her child’s appointment one week resulting in the child not being present at all. This was mirrored back by the other participants’ screen, as another mother, usually very attentive to her child, became absorbed in the adult conversation leaving her child to wander off and out of the screen altogether.

These events were unpacked in supervision, exploring what parallel processes were impacting facilitators own capacities to keep the infants in focus. The pleasure of seeing these two mothers so intimately connecting exceeded their desire to keep the group grounded from the infant up, whilst simultaneously leaving them feeling powerless to stop this. This provided a profound insight into what the infant themselves may feel, and an insight into perhaps something these mothers have felt in past relationships. The ability to hold both in mind, themselves (the adult), and their infant, was gently explored and articulated in that weeks’ newsletter.

In subsequent groups facilitators saw a marked shift in mothers directly talking to their child about the other children, and the children directly talking to other children and the facilitators onscreen. Objects and activities which had been previously sent out to the families were making reappearances during these sessions, with the children finding pleasure in the repetition of having certain books and games played. Group songs, accompanied by the time lag inherent in using online technology produced much laughter. To the delight of the children an unexpected visitor, the pet of one of the facilitators, made an appearance and captured their imaginations. This led to one child then bringing their plastic animals to the screen to introduce to the group.

Reading books to the children proved to be a powerful activity to bring the focus of the group back to the children. Facilitators were able to share a book that was of cultural relevance to the group members, and which then became a favorite. A regular ritual of the group, eating morning tea together (albeit it separately) became an activity that began to transcend the boundaries of these virtual worlds as the children were offered and pretended to take food from others in the group. On occasion the children themselves would take up the whole screen, appearing to move so close as to look more fully into the worlds’ of the other group members.

### 2.3. Service Two—Group One Online

The facilitation team in this second service consisted of a female and male facilitator. Whilst three families were to participate into their first online group only one young mother with her two infant children materialized. The two facilitators were left with a dilemma of continuing with only one family present. This family wanted the sessions to continue and halfway through the course of the eight sessions they were joined by another group member, the young child of the mother’s new partner.

With less faces needing monitoring on screen these two facilitators felt an immediate connection with the young children in the family. The older sibling noticed an animal toy in the background of one facilitators room and asked if they could have it. This exchange was managed playfully by facilitators morphing the toy (Peter the Pet Pig) into a mascot who appeared each week dressed to match the activities and elements which were being played with each week. Symbolically, this toy represented how the facilitators kept the family in mind between groups, preparing the pig in readiness for each ensuing group.

The facilitators observed quite different relational dynamics between the two children and their mother and wondered if the male child triggered responses within this mother which aligned his behavior with the violence used by the children’s father, with whom they no longer had contact. This little boy demonstrated great sensitivity and was encouraged by his mother to play dress ups and engage with a range of gender diverse toys. He could, however, get cross and his outbursts, comparable to children his age were interpreted by his mother as abusive. His little sister was torn between wanting to support him, even when he was cross, and accepting her mother’s invitation to admonish her brother’s behavior. This relational triangle was stretched even further when the stepdaughter, the same age as the little boy came to live with the family for several weeks.

This third child demonstrated a great deal of curiosity about the two facilitators on screen and dominated the space, demanding this mothers’ attention, and showing little interest in the other children. The facilitators felt a pervading sense of sadness during this session and noticed the little boy becoming more withdrawn. He had a physical altercation with both girls during the session and removed himself away from the center of the room as though feeling ashamed. The presence of the third child seemed to accentuate the relational ruptures already at play in this little family unit and offered facilitators a stark insight into the everyday world of these very young children.

Supervision provided the space within which these two facilitators could sit with their own feelings of impotence. They grappled with both the rawness of what they witnessed through their screens, resisting judgement, and embracing curiosity about what each child may have been experiencing. They also saw the struggle this young mother had attempting to emotionally embrace each child whilst trying to redress past feelings of powerlessness she had experienced at the hands of her children’s father. As they unpacked their own feelings in supervision, and the discomfort they felt in what they feared was them judging this mother, they were able to sit with their own sense of powerlessness. This in turn shifted how they then engaged with the emotional rawness they felt within each session and the comfort they were able to offer each child, the mother, and their evolving relationships.

A space opened within the subsequent sessions where facilitators were able to give voice to and be curious about what it may feel like to be each child and the big emotions they may be struggling with as they sought to have their needs met. This included exploring how this mother had her needs met when she was a child and the enormity of the damage she experienced within her intimate relationship with the children’s father, the violence the children had witnessed and the incubation of fear that surrounded their beginnings as a little family unit. Over the remaining weeks external pressures wrought by legal matters connected to the past family violence and current custody issues surrounding the third child left the mother exhausted. However, she and the children remained steadfast in their attendance and appeared to find great comfort in the emotional containment these weekly sessions offered them, giving facilitators feedback about how helpful she found the observations they shared with about her and the children, and their focus on giving herself and the children time to heal.

### 2.4. Service Two—Group Two Online

In contrast to this service’s first, ‘one family only’ online group, three toddlers and three older mothers participated in this group. The participants in the group identified themselves and/or the father of the children as Indigenous Australian or Anglo Australian. This made for a busy screen and facilitators worried that children would disappear from their screens and their minds. One mother who attended the online group presented as almost disembodied in early sessions. Her child frequently came up to the screen with fluffy toys and other important items, however, she struggled to see these opportunities for play and interaction, unable to respond to them. The facilitators were cognisant of a helpless, passive feeling of something not coming to fruition as they watched this toddlers’ overtures.

This dyads journey through the group was striking. The repetition of this toddler approaching and being rebuffed whilst the mother continued to interact with the group online angered one of the facilitators. In supervision the facilitators were supported to understand that the infant was likely to also feel very cross with her mother but at such a young age possibly overwhelmed by such a strong emotion and likely to internalize this rejection. Making these complex feelings overt shifted the anger felt by the facilitator as did an emerging and fuller picture of this mother’s own personal struggles. The transformation of the feelings evoked within this facilitator about this dyad, and how they then made themselves available to this mother, was matched by a transformation within the dyad’s relationship itself.

In subsequent sessions facilitators observed the mother engaging in play with her child, reflecting on her relationship with the child and most profoundly, turning away from facilitators and focusing on her child when her child asked for her attention. Facilitators were aware of a marked change within their own bodies when observing this dyad, conscious of how calming and soothing it felt to witness this reparation in their relationship. This was powerfully illustrated by an exchange which occurred where the mother was describing to facilitators something she had purchased for her child when the child suddenly disappeared from the screen. Having witnessed this child disappear so many times, the facilitators felt some trepidation. However, the child soon reappeared holding the purchase to show the facilitators what her mother had just told them about. This young child had been listening to the conversation and understood she was being spoken about and found a way to join the conversation, understanding that the group, albeit online, was her group as much as her mother’s.

A different dyad initially shared delightful reciprocity in the early weeks, but when issues relating to the child’s access with his father arose, past traumas for both mother and child were reactivated. The facilitators were required to change their expectations of this dyad and recognize that the sudden change in their presentation in the group space gave permission for the entire group to sit with the distress and ambivalence inherent in a familial space where violence has occurred. On one occasion the toddler in this dyad sat in the mother’s lap and pretended to be a baby again whilst this mother joined in this play. The facilitators wondered aloud what this was like for the rest of the group to watch this intimacy on their screens.

The third dyad was required to move back into the infant’s grandparents’ home until the mother could secure her own housing. This mother was observed to have a tenuous connection with her toddler and despite the violence she had experienced at the hands of the child’s father, felt he was a wonderful parent. As occurs within face-to-face groups, some women retain an enduring love for their partner, an ambivalence towards them or attempt to completely banish them from their lives. The ghosts of the father’s presence haunts the group space [4,15]. This was no different in this online space as the role, relationship with and ethereal presence of the fathers can be a point around which the women bond or splinter as a group. Further to this, it became apparent to facilitators in an early session that this mother’s stepfather was present in an early session but was just out of screen. Whilst the presence of an uninvited male in the group, along with this mother’s regard for her ex-partner could have splintered the group, that the male facilitator made transparent the presence of both the stepfather and the ambivalence women hold about fathers, managing to contain and make available the topic of men’s violence and fathering as an issue to be safely explored in the group space.

## 3. Discussion

The global experience of the COVID-19 pandemic has opened virtual pathways of relational connectivity previously unexplored. Simultaneously, a collective level of social anxiety and social isolation has increased in ways not yet fully understood [16]. Within Australia, and Victoria in particular, the result of imposed social and geographical lock downs has resulted in countless infants being deprived of forming important and critical early attachments with grandparents and extended family members, as well as socializing with other infants. Their parents had been unable to access to imperative, and much relied upon familial and community supports. Just how much this period has exacerbated existing family violence or triggered the commencement of relational volatility is beginning to emerge [36].

These four-separate online (8 to 10 week) group work experiences were offered to ensure that supportive therapeutic interventions were made available to vulnerable families during a time when levels of risk were exacerbated by the imposed isolation and stress of the pandemic. The ability to provide good therapeutic practice through a medium largely underemployed by support services, in highly populated cities or towns at least, has opened future opportunities to offer remote areas, or socially isolated families a viable alternative to face-to-face participation in group work programs. This exploration of what happened during a period of sequential lockdowns in two different states of Australia has yielded some interesting practice learnings for the facilitators involved and are worthy of further consideration.

### 3.1. Safety

Family violence often occurs behind closed doors. Virtual groupwork offered some insight into what was behind the closed doors of the homes of the families we worked with. On three sperate occasions, in different groups, facilitators became aware that there were other people present in the online sessions without this being made explicit by mothers. One incident involved a mother taking several phone calls from a partner who was court ordered to desist from making any contact with the mother or child. Remaining in contact with, or resuming contact with partners who are violent, despite legal rulings, is far from uncommon in family violence work [22]. Just who the phone calls were from did not become apparent immediately, but facilitators were able to follow this up outside of the group time and explore what this contact may mean for both the mother and the infant.

Another two separate incidents involved facilitators becoming aware of other people being present in the house (one of whom was the perpetrating partner) but who were not overtly identified by the mother as being present. Again, facilitators were able to reach out to families outside of the group time to offer support and assistance to ensure that the infant and mother were safe, either directly themselves or through existing connections the family had with other workers within their agency. Clear and informed consent had been gained by facilitators before commencing each group, where participants agreed to ensure the confidentiality of others, not to record sessions and clarity about who would be present in the group. This was in addition to the general PABC forms which clearly inform the mother that facilitators will discuss any issues of concern regarding the infant/child with them directly and will contact child protection if there are any risks regarding their safety.

Therapeutic work with infants exposed to violence requires vigilance in protecting the emotional and physical safety of the child. The instances described above raised concerns about the safety of both infants/children and their mothers with regard to others, something that would not have been apparent in office-based work unless disclosed by the mother or child. This new, virtual way of working provided weekly contact directly into the family’s homes. Information about the infant’s emotional world and the relational dynamics between mother and infant/child were revealed in a manner quite different to what routinely occurs within face-to-face office-based work. For example, the infant/child in their own home environment was more reliant on their engagement with their mother than what may occur when you have a room full of infants, mothers, and facilitators. It has not been unusual to see infants seeking out other adults over their mother. The busyness of the interactions in the group room space can sometimes obscure or delay the emergence of a coherent picture of the relational dynamic between infant and mother. The mother and infants/children, in their own home environment, offered somewhat greater clarity around how their relationship with each other worked, and just how they made use of each other in their own personal environment. In some instances, the impoverishment in the dyad’s relationship was starkly apparent.

In a group run by Service One, a dyad, who often commenced the group whilst still in bed, presented with a detached, rather avoidant relationship. The screen was often turned off whilst the mother dressed and tidied the hair of her toddler, however, the sounds of the toddler’s distress whilst having her hair brushed and tidied was clearly heard and vicariously felt, by the whole group. When their screen was on, the same toddler was often seen immersed in her iPad, initially showing little interest in other group members with the mother more intent on engaging with other mothers than her child. Concerns about this little one’s emotional safety within the relationship with the mother were canvassed early in the weekly supervision sessions, as was reflecting on ways in which facilitators could invite this little toddler and mother to join together in the screen space.

In a group run by Service Two, the sudden appearance of a mother’s new partner’s young child in the group halfway through the eight-week program called for facilitators to have to quickly adjust and adapt to include this new little group member. This child’s presence appeared to heighten the frailties in the mother’s relationship with her own children. Facilitators were able to use this shift in the dynamics between this little family unit to illuminate and gently explore some of the displaced feelings the mother carried about her ex-partner and appeared to project onto the child of this man who had been so violent towards her. This unexpected inclusion was illustrative of the flavor of unpredictably that occurred across all the groups. Office based group work allows facilitators much greater control of the environment. This direct window into the homes of the families offered a perspective which was in some measure more honest and in some ways more brutal.

### 3.2. Voyeurism and Entitlement

Whilst home visits are common practice in many services, offering support to families, the use of the online space carried a distinctly different quality to that of face-to-face work in a family’s home. The reason for using the online medium was because the families as well as the facilitators were in lock down. This meant that whilst facilitators could see into the home of the family, the family could also see into the home of the facilitator. Facilitators grappled with feelings of intrusiveness on occasions. This was in part associated with the instances where it became apparent that there were others in their home during the online sessions, yet this was not disclosed to facilitators. The bind for facilitators was respecting the privacy and right of the family to have what happens in their home kept as their own private business, with, their concerns about the safety of the children. Additionally, the rawness of some exchanges between the mother and their child, perhaps unfiltered by the fact that they were in their own homes, was confronting for facilitators on occasion. Sitting with these complex feelings and dynamics, unpacking these in supervision and offering the infant/child and mother containment through the online space was challenging.

Seeing into the homes of the group participants, the participants seeing into the homes of each other, and seeing into the facilitator’s homes raised discomfort and concerns about entitlement for some facilitators. Some participants had very few possessions in their homes and lived in cramped or dilapidated accommodation. Much of their homes were shown onscreen as they followed their child/children around with iPad or mobile phone. In contrast, facilitators tended to remain in one spot, and initially some covered up their background to avoid disclosing too much personal information about themselves. However, over time this was not always possible and background screens were problematic, impacting the use of toys and books, as the screen would obscure a clear vision of such objects. Facilitators felt a sharp sense of the disparity between the homes they lived in with the homes the participants lived in. The contrast between what possessions and choices the facilitators had at their disposal with that of the participants was keenly felt.

### 3.3. Restorative Play

Crucial to work with all ages, but particularly so in infant mental health work, is the capacity for reciprocity and play. Experiences of intergenerational trauma and family violence are not conducive to encouraging playfulness. The capacity to explore, imagine and play is imperative for the development of introspection and the ability to make conscious our internal world [37]. The ability for healthy and restorative play ideally forms in early childhood [37,38,39]. Where the mothers in this group had either experienced few opportunities for play in their own childhood or were denied an environment which enabled safe exploration and play, their infants and young children exhibited restricted play. Despite an initial apprehension felt by facilitators about playing in the online space their fears quickly abated.

The presence of Peter the Pig, present in every session with costume changes to reflect each week’s changing activities, was a lightning rod for creativity and play. Along with the children, facilitators were able to co-create a story where Peter became an imaginative and highly symbolic member of the group. Facilitators found that when they sang nursery rhymes together with the infants/children and mothers, the lag in time created by each separate online space meant that the songs often sounded out of kilter and rather garbled. This led to much merriment with facilitators able to enjoy, laugh and embrace this menagerie of ‘out of tune’ singing which encouraged the infants/children and mothers to do likewise. The energy created by the playfulness of the facilitators was contagious. Amongst what were at times very painful discussions, observed dynamics and reflections, the group space was tempered with generous amounts of restorative playfulness so important to bringing healing and reparation.

### 3.4. Virtual Relating

Prior to the pandemic the world globally has become increasingly dependent on social media and relating through online mediums [40]. COVID-19 has further hastened the dominance of this virtual means of relating to others. Just what did the infant/child make of the online group experience? What facilitators observed was that the infants and children increasingly sought out interactions with them through their screens, often coming right up to the screens as if looking through a window to get a better look at them and trying to peer into their homes. The small children were able to remember activities and books from previous sessions and asked for these to be reread or songs sung again.

Following the cessation of the lock down in one state, facilitators organized a face-to-face reunion with their online PABC group. The young children immediately recognized the facilitators and each other, and these children were quick to engage with one another, take control of the group space and create games together. Such had been the impact of the group that one of the grandmothers of a toddler in the group made a special visit towards the end of the reunion as she wanted to meet the facilitators. Where one family was unable to make it to the reunion because of health concerns, facilitators dropped off a parcel of food and treats to their house. The sense from the facilitators was that the regular weekly commitment of the online group was eagerly anticipated by the children and that they, and their mothers, were able to make good use of the online space. The online group became an important relational connection during a time where most felt extremely isolated. The children recognized that this was a time and space where they and their mothers spent special time together and mothers reported that the activities enjoyed online in the group became something they would do together in their homes outside of group sessions.

### 3.5. Accessibility

An important outcome of using these online platforms was the accessibility of providing services to sometimes hard to engage, hard to reach and occasionally chaotic families who find it hard with very young children to get to appointments. Not all families have access to the use of a car or the resources to pay for or manage using public transport. A subsequent group run by one of the service’s was permitted to run a face-to-face group only to reach week four and a state-wide five-day lockdown was announced. The fifth week was run online and then the remaining sessions were again run face-to-face. The capacity for providing a hybrid of online and face-to-face services to very vulnerable families has opened invaluable possibilities for improving the reach of service provision in ways not previously utilized nor imagined.

### 3.6. Kept in Mind

At a time when many struggled with isolation and loneliness due to the pandemic all facilitators went to extraordinary lengths to keep these families in mind. Over and above the weekly session’s facilitators dropped off goods and activities to the front doors of these families. They made phone contact and ensured that the weekly therapeutic newsletters were delivered to each family before the next session. One facilitator drew a beautiful portrait of a family and dropped this off framed at the front door to commemorate their time in the PABC online. Culturally appropriate books were sourced by the facilitators of one group and delivered to their door so that the children could enjoy seeing pictures of children and families in a story book that looked like them. At a time when so many vulnerable families were hidden away, the facilitators of these groups kept their focus on these infants/children and mothers at the forefront of their minds.

### 3.7. Inclusion of Male Facilitators

Involving men in the facilitation of the groupwork programs for children (aged 8 to 12) and mothers, as well as involving fathers post attending men’s behaviour change programs, ref. [26] led to the development of the PABC and a subsequent infant/father intervention [41]. This previous work had long taught us that the inclusion of male facilitators contributes to both our male and female infants and toddlers and parents; expanding their capacity to see others beyond their gender. This impacted the nuance of our work with children, “as we began seeing their family members as people, not genders, they showed themselves as the complex, ambiguous, uncertain and confused young people they were” ([22], p. 32).

Experienced, gentle and nonviolent men who have been part of the facilitation team for the PABC groups have offered the participants validation that men who use violence are not entitled to do so and challenged a sometimes-internalized belief that as women and infants/children they in some ways were responsible for or deserving of the violence they were subjected to. The healing power of using a female/male facilitation of the two groups run within one of the Australian states described in this paper proved to be no exception.

### 3.8. Strengths and Limitations of This Case Report

The virtual nature of the four online group work interventions described in this paper produced neither a better nor worse outcome for participants than what facilitators felt occurred in their face-to-face work. What facilitators found was that online work involved a different interactional process. This called on the facilitators to quickly adapt to emerging challenges and to use their weekly online supervision space to more comprehensively unpack events which occurred and sit with the uncertainties this new way of working generated for all. Facilitators felt the holding presence they provided was sometimes compromised, as painful or complex dynamics occurring between the mother/infant/child occurred remotely, occasionally thwarting the capacity to sit with, reflect on and hold in the space explorations into what the infant/child might be expressing to their most significant attachment figures.

Additionally, screens could readily be turned off and on. When this occurred, it was difficult to ascertain if this was deliberate or a technical difficulty. Returning after being absent from the screen impacted the flow of the group for those infants who were mobile as it was dependent on the mother to either follow them on screen or keep them engaged in the space. This risked disappearing the infant visually and mentally. Where triggering conversations, events or interactions occurred online it was difficult to check in with individual dyads as could occur when meeting in person. Online sessions ended suddenly with participants clicking out at the conclusion of the group inhibiting the opportunity to check in with individual families as would normally occur in our face-to-face work, and as they were packing up to leave the in-person sessions.

Qualitative feedback from participants, and extremely high attendance rates clearly indicated that their participation in the PABC online helped ameliorate the isolation created by lockdown measures. Children’s activity packs delivered to their homes enabled the children to feel connected to other children online and cemented a day and time where the children became excited as they knew it was time for their special group. Facilitators felt the online sessions provided them with a transparency about the lives and issues faced by these families which office-based work kept hidden. The online space also extracted a greater flexibility and creativity from the facilitators themselves and provided a vitally safe, consistent space for these families to be seen, heard, and kept in mind. The challenges in keeping the infant at the fore refined their capacity to bring the subjectivity of the infant central to the work as the infants and young children moved in and out of the screen space.

## 4. Conclusions

Lockdowns continued intermittently within both these states within Australia until the end of 2021. The dexterity with which these practitioners used online platforms to deliver therapeutic treatment has grown significantly. Just as technology and the immense increase in infants and children’s exposure to screentime is a fact of life, so too is COVID and future risks of viral implosions across the globe. How to use technology to enhance relational opportunities, to bring repair and constancy to the lives of the rapidly developing infant and young child is imperative if the therapeutic community is to reach and treat families at risk. Whilst many services very successfully offer home based support, many practitioners in mental health, private practice and community health are reliant on office-based services. For extremely vulnerable families, great flexibility in the provision of services, and a virtual reaching into the homes of infants and young children may offer a viable, and as powerful transformative experience, as does face-to-face work.

## Data Availability

Not applicable.

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
