# Peer review of "Once upon a Pandemic: ‘Online’ Therapeutic Groupwork for Infants and Mothers Impacted by Family Violence"

_ijerph, 2022, doi:10.3390/ijerph192316143_

Round 1
Reviewer 1 Report
I'm very conflicted about the article because I share the experience of being in the clinical practice dealing with the reality of doing, and from this point of view, I found very valuable the recount of the experience of the authors. Clearly, the work with the group with Mother- children dyads during the pandemic is praiseworthy. On the other hand, in terms of qualitative research methodology, there are many limitations in this article. I recommend that it should be presented (and reviewed) as a case report. In those terms, the authors should describe more precisely all the formalities of the procedure, for instance, how they determined the composition of the groups, and what are the criteria or estimation of the results of intervention, in terms of the feedback with the dyads (subjective), in terms of the researcher (subjective) and in terms of any quantitative measure ( objective) if there is any ( I'm not placing numbers above subjective experience, I'm just suggesting a way to systematize the results report). That could also improve the discussion and conclusions, adding new questions and possibilities to design further research for the future.
Author Response
Thank you for your thoughtful feedback. We very much appreciate your honesty. We are aware our methodology section is weak as the circumstances surrounding the lock downs threw all our normal processes out of kilter. We also decided as we went along to compare notes between the two different states who had moved online so we could learn from each other as we slowly found our respective new ways of working ‘online’ as opposed to face to face. We (the facilitators from each state) presented an online paper at our national infant mental health conference on what we had learnt through this process and because of the level of interest in what we had done we decided to write up our experience. We have shortened the length of the paper and re-presented this new paper as what it really is, a case study rather than a solid evaluation.
Reviewer 2 Report
In my opinion authors made a deep description of the experience of delivering the group intervention online, but it is a description of a process and not a research article. The content included in the different sections do not correspond to what should be there: there is no reference to research design, methods, results'analysis, etc. The discussion includes implications for practice, not discussion of the results of a research in face of the literature reviwed and the actual state of art.
This could be an interesting article about the differences between online and in face results of PACB group intervention, but in the present form I think it is a rich and useful description for professionals but it is not a research article.
Author Response
Thank you for your review. We agree with your conclusion. It is not a research article, nor did we ever really plan to do this intervention and provide a thorough evaluation as circumstances threw us into a space where we had to suddenly change the way we did everything, something in itself which we felt made this piece of work interesting and worth sharing. We have removed any attempts of attempting to present this as a research article and represented it as a case study, offering as you suggested, “a rich and useful description for professionals”.
Round 2
Reviewer 1 Report
Unfortunately I've found very few changes in the new ersion of the paper, and it doesn't reach the level of quality required to be published.
Author Response
Thank you for your comments, we have revised our paper according to the reviewer's and academic editor's comments.
Reviewer 2 Report
In my opiniom this is a relevant and useful analysis of a case report about an intervention delivered online due to pandemic situation. The change of perspective (from original research article to case report) and the changes introduced improved the article , giving a relevant contribute for professionals practice.
Author Response
Thank you very much for your review and comments.